# MultiColor: Image Colorization by Learning from Multiple Color Spaces

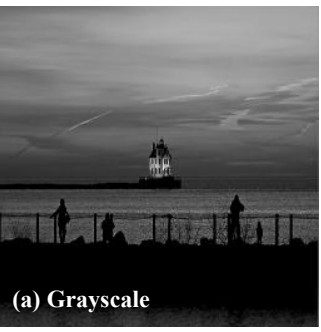 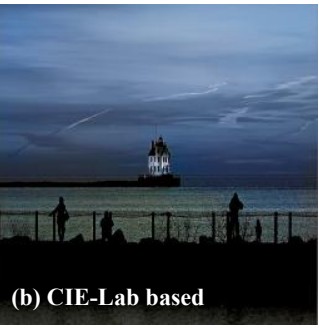 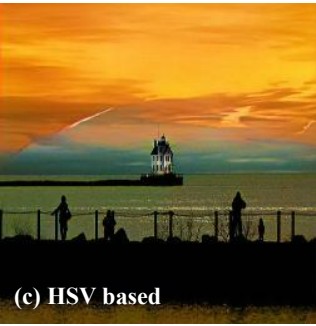 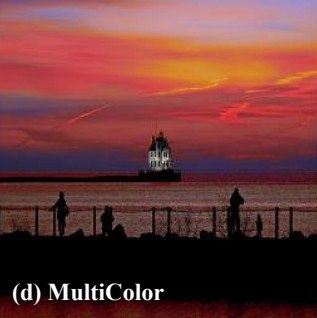

(a) Grayscale    (b) CIE-Lab based    (c) HSV based    (d) MultiColor

Figure 1: Image colorization with MultiColor. Given a grayscale image (a), colorization results with single color space (b,c) may be bias, and our approach learns from multiple color spaces to obtain colorized image with more detailed color information (d).

## ABSTRACT

Deep networks have shown impressive performance in the image restoration tasks, such as image colorization. However, we find that previous approaches rely on the digital representation from single color model with a specific mapping function, a.k.a., color space, during the colorization pipeline. In this paper, we first investigate the modeling of different color spaces, and find each of them exhibiting distinctive characteristics with unique distribution of colors. The complementarity among multiple color spaces leads to benefits for the image colorization task.

We present MultiColor, a new learning-based approach to automatically colorize grayscale images that combines clues from multiple color spaces. Specifically, we employ a set of dedicated colorization modules for individual color space. Within each module, a transformer decoder is first employed to refine color query embeddings and then a color mapper produces color channel prediction using the embeddings and semantic features. With these predicted color channels representing various color spaces, a complementary network is designed to exploit the complementarity and generate pleasing and reasonable colorized images. We conduct extensive experiments on real-world datasets, and the results demonstrate superior performance over the state-of-the-arts. The code will be available.

## CCS CONCEPTS

• **Computing methodologies**; • **Computer graphics**; • **Image manipulation**; • **Image processing**;

## KEYWORDS

Image colorization, multiple color spaces, complementary network, visual transformer

## 1 INTRODUCTION

Image colorization is an artful process that seeks to infuse grayscale images with color in a remarkably realistic manner [5, 8, 40, 45]. This technology is pivotal in various fields, including digital art enhancement [33] and legacy photos/videos restoration [41, 42, 47]. The core challenge is how to predict the missing color channels that achieve satisfactory image colorization. Overcoming the challenge requires leveraging potential models that can resolve the interactions between color channels and grayscale.

Over the last few years, deep learning techniques have yielded significant improvements in image colorization. Early methods [8, 36, 46] employ convolutional networks to predict the per-pixel color distributions, while recent works [18, 21, 43, 44] take advantage of pre-trained generative adversarial networks [10] or transformer [38] to pursue the vividness and fidelity of colorized images. The pixels within the color image can be represented in various color spaces, such as RGB, HSV, and CIE-Lab. Existing image colorization approaches usually train the model under a specific color space. As shown in Figure 1(b,c), we observe that they may suffered colorized results bias, especially for the contents with complex color interactions. The color spaces are designed to support the reproducible representations of color with unique properties, and incorporating the properties across different color spaces is helpful to generate pleasing and reasonable colorized images (Figure 1(d)).

In this paper, we develop a learning-based approach, namely MultiColor, that utilizes the complementarity from multiple color spaces for image colorization. Starting from an encoder that extracts

multi-scale feature maps, we employ a set of dedicated colorization modules for individual color space. Inspired by the success of query-based methods [4, 6, 7, 18], the colorization module for a specific color space also employs the powerful attention mechanism combined with a sequence of learnable queries. Specifically, a transformer decoder is first employed to refine color query embeddings, and then a color mapper produces color channel prediction using the embeddings and semantic features. In order to fully tap the potential of multiple color spaces, we further introduce the color space complementary network. The contributions of this paper include the following:

- Unlike the existing works based on single color space, the proposed MultiColor employs learnable color queries to modeling various color channels of multiple color spaces, which are utilized to generate visually reasonable colorized images. To our knowledge, this is the first work to image colorization with multiple color spaces.
- We design a simple yet effective color space complementary network to combine various color information from multiple color spaces, which helps maintain color balance and consistency for the overall colorized image.
- We validate MultiColor through extensive experiments on the ImageNet [34] colorization benchmark, and demonstrate that MultiColor outperforms recent state-of-the-arts. Furthermore, we test our model on two additional datasets (COCO-Stuff [3] and ADE20K [48]) without finetuning, and our approach shows very competitive results with multiple color spaces.

The remainder of this paper is organized as follows. Section 2 discusses related works and Section 3 briefly reviews the background of our work. Our proposed MultiColor is described in Section 4 and extensively evaluated in image colorization experiments in Section 5. Finally, Section 6 concludes this paper.

## 2 RELATED WORK

**Image Colorization** aims to add color information to a grayscale image in a realistic way. Cheng *et al.* [8] propose the first deep learning based image colorization method. Zhang *et al.* [46] takes grayscale image as input and predicts the corresponding ab color channels of the image in the CIE-Lab color space. InstColor [36] presents an approach to colorizing grayscale images that takes into account the objects and their semantic context in the image. Some researchers introduce rich representations from pre-trained GAN [2, 19] into the colorization task. Besides, an adversarial learning colorization approach is employed to infer the chromaticity of a given grayscale image conditioned to semantic clues [39]. GCPColor [44] produces vivid and diverse colorization results by leveraging multi-resolution GAN features. BigColor [21] utilizes color prior for images with complex structures. Despite the expansive representation space afforded by GANs, these methods encounter several constraints, particularly when dealing with images exhibiting complex structures, resulting in inconsistent results. Recently, Transformer [38] has been extended to image colorization task [17, 18, 24, 43]. ColorFormer [17] utilizes a Color Memory Assisted Hybrid-Attention Transformer (CMHAT) to generate high-quality colorized images. The key innovation of CT2 [43] is using

color tokens, which are special tokens that are used to provide information about the color palette of the image. DDColor [18] includes a multi-scale image decoder and a transformer-based color decoder. The two decoder aims to learn semantic-aware color embedding and optimize color queries.

Current image colorization techniques are primarily based on single color space which suffers from poor generalization ability. For instance, the models of [17, 18, 43, 44] are built upon CIE-Lab color space, and [8] employ the YUV color space. Although single color space has the ability to represent image color information, there are limitations imposed by the complexity of real-world scenarios, where color distribution can be more intricate and varied. To address the limitations, this work aims to integrate multiple color spaces, which can provide a more comprehensive representation of color information, adaptable to different contexts. By leveraging the diversity and richness afforded by multiple color spaces, we can effectively broaden color scope, offering more adaptable and flexible strategy for the colorization process.

**Color Space Combination.** The combined use of multiple color spaces has garnered significant attention in the computer vision field due to its ability to capture features at various levels. Color-Net [11] is introduced to demonstrate the significant impact of color spaces on image classification accuracy. Kumar *et al.* [23] proposed a method for enhancing foggy images by fusing modifications of image histograms in the RGB and HSV color space which significantly improves image contrast and reduces noise. Ucolor [26] enriches the diversity of feature representations by incorporating the characteristics of different color spaces into a unified structure. Peng *et al.* [27] fused multiple algorithms in RGB and HSV color spaces to improve brightness, contrast, and preserve rich details. Wan *et al.* [40] leveraged the RGB color space for the initial colorization of super-pixels and subsequently utilized the YUV color space for color propagation. This approach struck a harmonious balance between efficiency and effectiveness in the image processing. Mast [25] investigated loss designation in different color spaces, revealing that the decorrelated color space can force models to learn more robust features. DucoNet [37] explored image harmonization in dual color spaces, supplementing entangled RGB features with disentangled L, a, b channel feature to alleviate the workload in the harmonization process.

## 3 BACKGROUND

Given a grayscale image $I_g \in \mathbb{R}^{H \times W \times 1}$ with height $H$ and width $W$, the colorization model aims to generate colorized image $\hat{I}_c \in \mathbb{R}^{H \times W \times 3}$. One popular paradigm is to predict missing color channels $\hat{y} \in \mathbb{R}^{H \times W \times 2}$ and then compute $\hat{I}_c$ through specific color space mapping function $\mathcal{F}$, *i.e.*,

$$\hat{I}_c = \mathcal{F}(I_g, \hat{y}). \tag{1}$$

Here $\hat{y}$ can be *ab* color channels from CIE-Lab, *HS* from HSV, or channels in other color space that represent color. Given the color space to learn missing colors, the transform function $\mathcal{F}$ is fixed. Thus, it can not be optimized along with the colorization network during training.

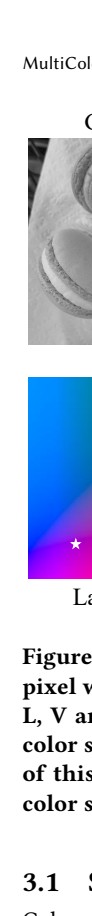

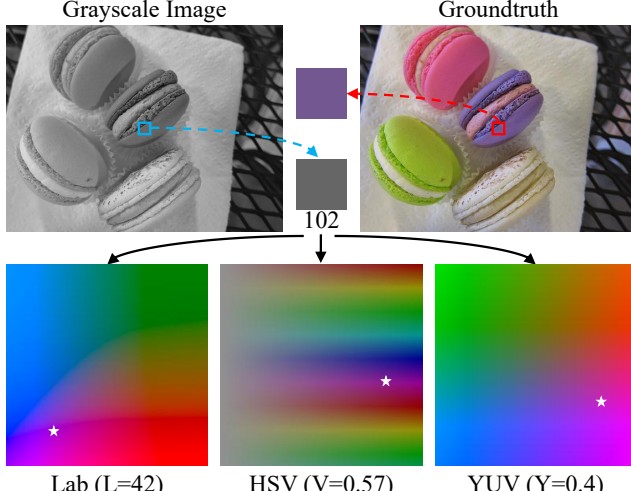

Grayscale Image          Groundtruth

102

Lab (L=42)     HSV (V=0.57)     YUV (Y=0.4)

**Figure 2: Color gamut of different color spaces at a specific pixel with grayscale value 102. The corresponding values of L, V and Y channels are 42, 0.57, and 0.4 in the respective color space. The pentagram indicates where the color value of this pixel in groundtruth color image appears in other color spaces.**

## 3.1 Suboptimal of Using Single Color Space

Color space refers to specific organization of colors which can be represented as tuples of numbers. Each of them offers a special perspective on color representation [9, 20, 31]. For instance, the CIE-Lab color space is designed to approximate human vision and perception which is device-independent and perceptually uniform, allowing for accurate color reproduction across different mediums and devices. The HSV color space is a cylindrical color model that represents colors based on hue, saturation, and value components representing image colors according to perceptual attributes. And the YUV color space separates the brightness from the color information components and is widely applied in video systems.

For the image colorization task, there are only the brightness values for each pixel in the input grayscale image. As various color spaces are utilized to define and reproduce colors in different digital imaging systems, they have with own distinct characteristics and limitations. The brightness can be mapped to one channel of the color spaces, such as the L channel in CIE-Lab and the V channel in HSV. Here we visualize the gamut of different color spaces at a specific brightness value in Figure 2. We can observe that the illustrated colors in different color spaces are inconsistent, and colors missing from one color space may appear in another color space. For example, the bright green color in the upper left corner of the YUV color space disappears in the other color spaces of the corresponding brightness values.

We believe that colorization learning in different color spaces will have different preferences. The channel values in different color spaces can be complementary, and utilizing information from multiple color spaces can boost the quality of generated colorized images. A variety of color spaces facilitate a more nuanced and comprehensive understanding of the color representation. The strategy

of multiple color spaces provides a wider range of color representations to accurately convey the subtle changes and complexity of colors in images. Combining the information from different spaces has the potential to overcome learning bias in single space. Through the utilization of multiple color spaces, we can harness their complementary attributes, facilitating a more comprehensive and nuanced depiction of color information. In the following section, we will present our approach to leverage the potential of multiple color spaces to colorize the grayscale images.

## 4 IMAGE COLORIZATION BY MULTICOLOR

Figure 3 illustrates the architecture of the MultiColor framework. It contains three components. The *encoder* aims to extract multi-scale feature maps that represent image contents. The *colorization modules* operate on the learned color queries and generate color channels for individual color space. Finally, the *color space complementary network* is utilized to combine the color channels of multiple color spaces to generate colored images. We describe below each individual component in detail.

## 4.1 Encoder

The encoder takes the grayscale image $I_g$ as input and estimates multi-scale semantic features, which are the essential foundation for the subsequent operations. We follow [18] and adopt ConvNeXt [28] for the feature extractor. The upsampling operation comprises four sequential stages which incrementally enlarge the spatial resolution of the features, and each stage is made up of an upsampling layer and a concatenation layer. Specifically, the upsampling layer is implemented through convolution and pixel-shuffle, while the concatenation layer also incorporates a convolution, integrating features from corresponding stages of the feature extractor by shortcut connections. The encoder generates 4 intermediate feature maps $F = \{F_1, F_2, F_3, F_4\}$ with resolutions of $\frac{H}{16} \times \frac{W}{16}$, $\frac{H}{8} \times \frac{W}{8}$, $\frac{H}{4} \times \frac{W}{4}$ and $H \times W$. Thanks to the downsampling and upsampling operations, our method can capture a complete feature pyramid. These multi-scale features are used as the input of the color space modeling stage to guide the optimization of the color query embeddings and color channels.

## 4.2 Modeling in Individual Color Space

As shown in the left of Figure 3, the colorization module contains two key parts, *i.e.*, the transformer decoder and the color mapper. The transformer decoder aims to refine $N$ learnable color queries through the multi-scale features. The color mapper predicts different color channels by receiving the refined color embedding and the feature map from the encoder. The whole pipeline follows the meta-architecture of Mask2Former [6].

**Transformer Decoder.** Standard transformer decoder [22] is employed to transform color queries by cross-attention and multi-head self-attention mechanisms. Color query embeddings ($\mathcal{X}_c \in \mathbb{R}^{N \times C}$) are refined by interacting with image features of resolution $\frac{H}{16} \times \frac{W}{16}$, $\frac{H}{8} \times \frac{W}{8}$ and $\frac{H}{4} \times \frac{W}{4}$. The transformer decoder can be formulated

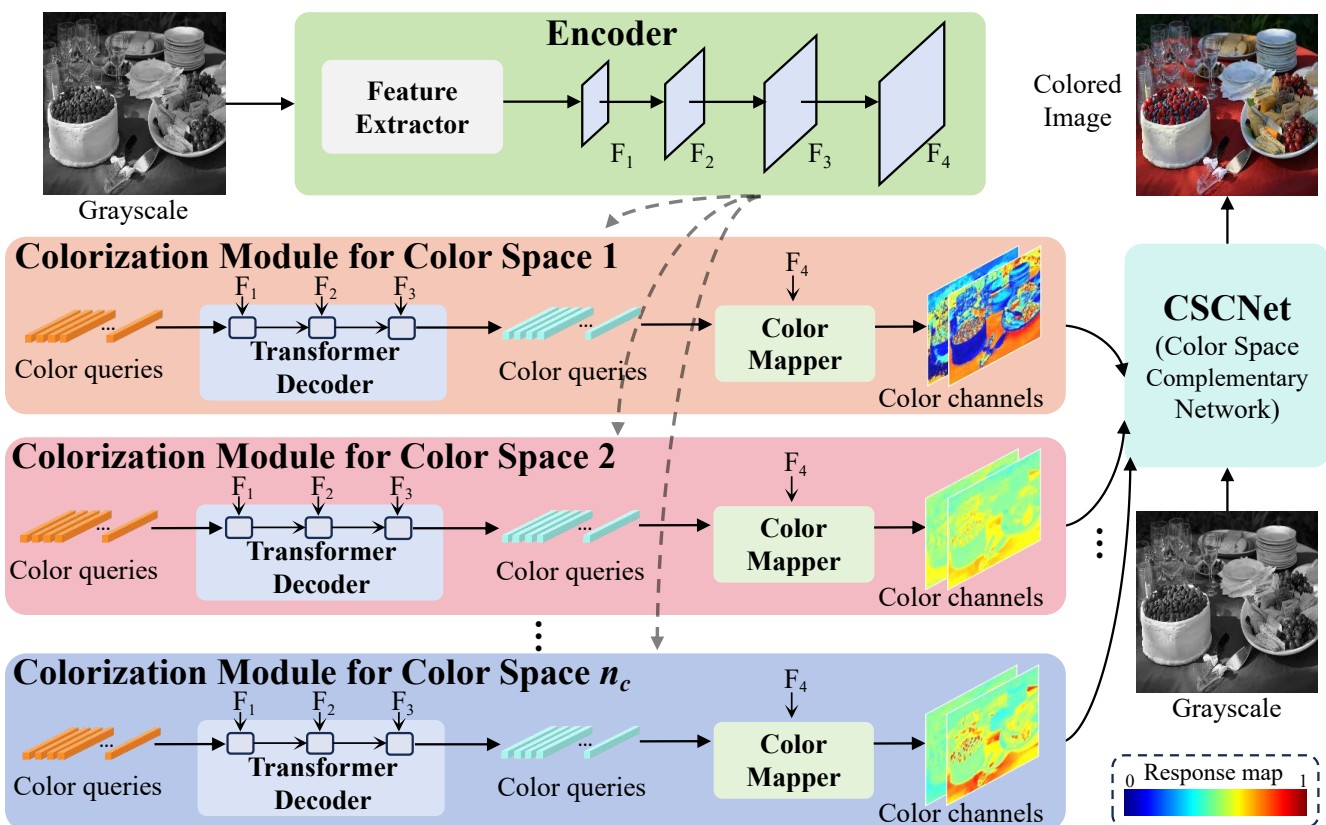

**Figure 3: The architecture of the proposed framework. Given a grayscale image, multi-scale semantic features are obtained with the encoder. Multiple modeling color space operations can produce various color channels of different color spaces. For each colorization module, transformer decoder refines learnable color queries based on the multi-scale features, and color mapper aims to generate color channels of multiple color spaces. Finally, the Color Space Complementary Network (CSCNet) is introduced to transform the multiple color channels into colorized images.**

as follows:

$$\mathcal{X}_c^{l'} = \mathsf{CA}(f_Q(\mathcal{X}_c^{l-1}), f_K(F_j), f_V(F_j)) + \mathcal{X}_c^{l-1} \quad (2)$$

$$\mathcal{X}_c^{l''} = \mathsf{MHSA}(\mathsf{LN}(\mathcal{X}_c^{l'})) + \mathcal{X}_c^{l'} \quad (3)$$

$$\mathcal{X}_c^l = \mathsf{LN}(\mathsf{FFN}(\mathsf{LN}(\mathcal{X}_c^{l''})) + \mathcal{X}_c^{l''}) \quad (4)$$

where $\mathcal{X}_c^l$ is color query embeddings at the $l^{\text{th}}$ layer, $F_j$ is the $j^{\text{th}}$ intermediate feature map. $\mathsf{CA}(\cdot)$, $\mathsf{MHSA}(\cdot)$, $\mathsf{FFN}(\cdot)$, $\mathsf{LN}(\cdot)$ indicate cross-attention, multi-head self-attention, feed-forward network, layer normalization; $f_Q$, $f_K$ and $f_V$ are linear transformations. We perform these sets of alternate operations $\frac{L}{3}$ times in the 3-layer transformer decoder. Specifically, the first three layers receive feature maps $F_1$, $F_2$ and $F_3$, respectively. This pattern is repeated for all following layers.

**Color Mapper.** The color mapper combines the feature map of the last upsampling layer in the encoder and the refined color embedding of the transformer decoder to produce color channels. The structure is shown in Figure 4, and it can be formulated as:

$$\hat{y} = \mathsf{Conv}(\mathcal{X}_c^L \cdot F_4) \quad (5)$$

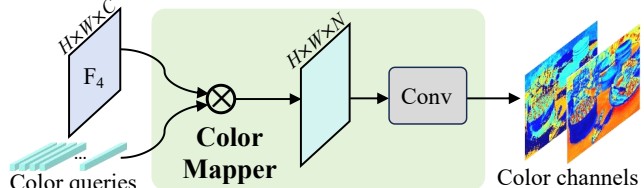

**Figure 4: The structure of color mapper.**

where $F_4 \in \mathbb{R}^{H \times W \times C}$ is the feature map, $\mathcal{X}_c^L \in \mathbb{R}^{N \times C}$ is the refined color embedding from the transformer decoder, $\mathsf{Conv}$ is $1 \times 1$ convolution layer, and $\hat{y} \in \mathbb{R}^{H \times W \times 2}$ is the predicted color channels from individual color space.

Since there are $n_c$ colorization modules representing multiple color spaces, we denote the predictions as $\{\hat{y}^1, \hat{y}^2, \cdots, \hat{y}^{n_c}\}$.

### 4.3 Color Space Complementary Network

The goal of CSCNet is to produce colorized image according to the brightness value and color channels. A straightforward approach

**Table 1: Configuration of the CSCNet. "conv$_k$, $c$" indicates the $k^{\text{th}}$ convolutional layer with $c$ output channels. By default $L_i = 1$.**

| Block1 | | Block2 | | Block3 | | Block4 | | Block5 |
|--------|--|--------|--|--------|--|--------|--|--------|
| conv$_1$, 32
BN+ReLU
conv$_2$, 32
BN+ReLU | $\times L_1$ | conv$_1$, 64
BN+ReLU
conv$_2$, 64
BN+ReLU | $\times L_2$ | conv$_1$, 128
BN+ReLU
conv$_2$, 128
BN+ReLU | $\times L_3$ | conv$_1$, 64
BN+ReLU
conv$_2$, 64
BN+ReLU | $\times L_4$ | conv$_1$, 32
BN+ReLU
conv$_2$, 3
BN+ReLU |

is to employ a transform function like Eq. (1). Here we employ an alternative approach by using a transform network $\Phi(\cdot)$, *i.e.*,

$$\hat{I}_c = \Phi(I_g, \hat{y}) \tag{6}$$

There are two benefits for this design. First, using the differentiable transform network keeps the model training in an end-to-end manner. Different from the fixed mapping function $\mathcal{F}$, the whole process with $\Phi$ is parameter-optimizable. Second, it is easier to extend under the multiple color spaces scenarios, as there exists no direct mapping function that converts multiple color space information into a single color space.

Given the predicted color channels $\{\hat{y}^1, \hat{y}^2, \cdots, \hat{y}^{n_c}\}$ from the colorization modules, the color space complementary network $\Phi_{csc}$ learn the color mapping from multi-color channels so that more color representation can be better explored for image colorization. The complementary network based on multiple color spaces can be formulated as:

$$\hat{I}_c = \Phi_{csc}(I_g, \hat{y}^1, \hat{y}^2, ..., \hat{y}^{n_c}) \tag{7}$$

The detailed structure of the CSCNet is listed in Table 1. The network is lightweight and consists of five blocks. Each block contains the convolution operation which is followed by batchnorm (BN) layer and rectified linear unit (ReLU) layer. Different settings of conv$_1$ and conv$_2$ will be examined in the ablation study. The scale of the output in each block is the same as the input feature. Note that, the output of block5 is 3, corresponding to the 3 channels of produced colorized image.

## 4.4 Training Objective

We adopt the following losses to train our image colorization network, including color channel loss, perceptual loss, adversarial loss, and colorfulness loss.

**Color Channel Loss** represents the low-level supervision of colors. Here $L_1$ loss is applied to perform color channel level supervision in predicted color channels and groundtruth color channels, *i.e.*,

$$\mathcal{L}_{cc} = \sum_i \lambda_c^i \left\| \hat{y}^i - y^i \right\|_1 \tag{8}$$

where $\hat{y}^i$ ($y^i$) and $\lambda_c^i$ are the predicted (groundtruth) color channels and weight for the $i^{\text{th}}$ color space.

**Perceptual Loss.** We adopt perceptual loss to capture high-level semantics and try to simulate human perception of image quality. We adopt a pre-trained VGG-16 [35] to extract features from both the colorized image $\hat{I}_c$ and the groundtruth color image $I_c$ and compute the loss as follow,

$$\mathcal{L}_{per} = \sum_j \lambda_j \left\| \phi_j(\hat{I}_c) - \phi_j(I_c) \right\|_1 \tag{9}$$

where $\phi_j(.)$ denotes the first convolutional layer of $j^{\text{th}}$ block of VGG-16 ($j = 1, 2, 3, 4, 5$). $\lambda_j$ is the weight of the corresponding layer.

**Adversarial Loss.** We also exploit the difference on the whole image level with the Adversarial Loss [10]. Specifically, we use the popular PatchGAN discriminator $D$ [16], and the adversarial loss $\mathcal{L}_{adv}$ is defined as follow:

$$\mathcal{L}_{adv} = \mathbb{E}_{I_c}[\log D(I_c)] + \mathbb{E}_{\hat{I}_c}[1 - \log D(\hat{I}_c)] \tag{10}$$

**Colorfulness Loss.** Following [18], we also introduce colorfulness loss $\mathcal{L}_c$ to generate more colorful and visually pleasing images:

$$\mathcal{L}_c = 1 - [\sigma_{rgyb}(\hat{I}_c) + 0.3 \cdot \mu_{rgyb}(\hat{I}_c)]/100 \tag{11}$$

where $\sigma_{rgyb}(\cdot)$ and $\mu_{rgyb}(\cdot)$ denote the standard deviation and mean value on the sRGB color space [12].

**Full Objective.** Therefore the full objective $\mathcal{L}$ for the image colorization network is formed as:

$$\mathcal{L} = \lambda_{cc}\mathcal{L}_{cc} + \lambda_{per}\mathcal{L}_{per} + \lambda_{adv}\mathcal{L}_{adv} + \lambda_c\mathcal{L}_c \tag{12}$$

where $\lambda_{cc}$, $\lambda_{per}$, $\lambda_{adv}$, and $\lambda_c$ are the trade-off parameters for different terms.

## 5 EXPERIMENTS

### 5.1 Experimental Setting

**Datasets.** We conduct experiments on three datasets: ImageNet [34], COCO-Stuff [3] and ADE20K [48]. We use the training part of ImageNet to train our method and evaluate it on the validation part (val50k). ImageNet (val5k) is the first 5k images of the validation set. Besides, in order to show the generalization of our method, we test on COCO-Stuff and ADE20K validation sets without any fine-tuning.

**Evaluation Metrics.** To comprehensively evaluate the performance of our method, we use Fréchet inception distance (FID) [14] and colorfulness score (CF) [12]. FID measures the distribution similarity between generated images and groundtruth images, and CF reflects the vividness of generated images. It is worth noting that a high colorfulness score does not always mean good visual quality, because it encourages rare colors, leading to unreal colorization results. Therefore, we provide the absolute CF difference ($\Delta$CF) between the colorized images and the groundtruth images. Besides, we provide Peak Signal-to-Noise Ratio (PSNR) [15] for reference, although it is a widely held view that the pixel-level metrics may not well reflect the actual colorization performance [13, 17, 30, 36, 39, 44].

**Table 2: Quantitative comparison of different methods on benchmark datasets. Best and second best results are in bold and underlined respectively. ↑ (↓) indicates higher (lower) is better.**

| Method | ImageNet (val5k) | | | | ImageNet (val50k) | | | | COCO-Stuff | | | | ADE20K | | | |
|---|---|---|---|---|---|---|---|---|---|---|---|---|---|---|---|---|
| | FID↓ | CF↑ | ΔCF↓ | PSNR↑ | FID↓ | CF↑ | ΔCF↓ | PSNR↑ | FID↓ | CF↑ | ΔCF↓ | PSNR↑ | FID↓ | CF↑ | ΔCF↓ | PSNR↑ |
| DeOldify [1] | 6.59 | 21.29 | 16.92 | 24.11 | 3.87 | 22.83 | 16.26 | 22.97 | 13.86 | 24.99 | 13.25 | 24.19 | 12.41 | 17.98 | 17.06 | 24.40 |
| CIC [46] | 8.72 | 31.60 | 6.61 | 22.64 | 19.17 | **43.92** | 4.83 | 20.86 | 27.88 | 33.84 | 4.40 | 22.73 | 15.31 | 31.92 | 3.12 | 23.14 |
| InstColor [36] | 8.06 | 24.87 | 13.34 | 23.28 | 7.36 | 27.05 | 12.04 | 22.91 | 13.09 | 27.45 | 10.79 | 23.38 | 15.44 | 23.54 | 11.50 | 24.27 |
| GCPColor [44] | 5.95 | 32.98 | 5.23 | 21.68 | 3.62 | 35.13 | 3.96 | 21.81 | 13.97 | 28.41 | 9.83 | 24.03 | 13.27 | 27.57 | 7.47 | 22.03 |
| ColTran [24] | 6.44 | 34.50 | 3.71 | 20.95 | 6.14 | 35.50 | 3.59 | 22.30 | 14.94 | 36.27 | 1.97 | 21.72 | 12.03 | 34.58 | 0.46 | 21.86 |
| CT2 [43] | 5.51 | 38.48 | 0.27 | 23.50 | 4.95 | 39.96 | 0.87 | 22.93 | 13.15 | 36.22 | 2.02 | 23.67 | 11.42 | **35.95** | 0.91 | 23.90 |
| BigColor [21] | 5.36 | **39.74** | 1.53 | 21.24 | 1.24 | 40.01 | 0.92 | 21.24 | 12.58 | 36.43 | 1.81 | 21.51 | 11.23 | 35.85 | 0.81 | 21.33 |
| ColorFormer [17] | 4.91 | 38.00 | 0.21 | 23.10 | 1.71 | 39.76 | 0.67 | 23.00 | 8.68 | 36.34 | 1.90 | 23.91 | 8.83 | 32.27 | 2.77 | 23.97 |
| DDColor [18] | 3.92 | 38.26 | 0.05 | 23.85 | 0.96 | 38.65 | 0.44 | 23.74 | 5.18 | **38.48** | 0.24 | 22.85 | 8.21 | 34.80 | 0.24 | 24.13 |
| MultiColor [ours] | **2.17** | 38.24 | **0.03** | **24.69** | **0.42** | 38.89 | **0.20** | **24.58** | **2.59** | 38.10 | **0.14** | **24.53** | **3.65** | 35.21 | **0.17** | **25.37** |

**Implementation Details.** In order to keep fair comparison with the sota methods, we follow some experimental settings with previous methods [17, 18]. Our method is implemented using PyTorch [32]. The network are trained from scrach with AdamW [29] optimizer and set $\beta_1 = 0.9$, $\beta_2 = 0.99$, weight decay = 0.01. For the upsampling stages in encoder, the feature dimensions are 512, 512, 256, and 256, respectively. We empirically set $N$=100 for the color queries in colorization modules. The hyper-parameter $\lambda_c^i$ controls the color information weight of different color spaces. We set them to 0.1 for CIE-Lab, 10 for HSV and YUV. Besides, we set $\lambda_j$ to 0.0625, 0.125, 0.25, 0.5 and 1 when $j = 1, 2, 3, 4, 5$, respectively. We set $\lambda_{cc}$ to 1.0, $\lambda_{per}$ to 5.0, $\lambda_{adv}$ to 1.0, and $\lambda_c$ to 0.5. The whole network is trained in an end-to-end self-supervised manner for 400,000 iterations with batch size of 16. The learning rate is initialized to $1e^{-4}$, which is decayed by 0.5 at 80,000 iterations and every 40,000 iterations thereafter. All the images are resized to $256 \times 256$. During training, we adopt color augmentation [18, 21] to real color images. The experiments are conducted on 4 Tesla A100 GPUs.

## 5.2 Comparison with Prior Arts

To evaluate the performance of our method, we compare our results with previous state-of-the-art image automatic colorization, including CNN-based methods (CIC [46], InstColor [36]), GAN-based methods (GCPColor [44], BigColor [21]), and transformer-based methods (ColTran [24], CT2 [43], ColorFormer [17], DDColor [18]).

**Quantitative Comparison.** Table 2 shows the quantitative comparisons. For the ImageNet [34], COCO-Stuff [3] and ADE20K [48] datasets, the results of previous methods are reported by [17, 18]. For the missing results of GCPColor [44], CT2 [43] and BigColor [21] in COCO-Stuff dataset, we show the results by running their official codes. On the ImageNet, our method achieves the lowest FID value, indicating its capability to produce high-quality and realistic colorization results. Besides, our method gains the lowest FID on the COCO-Stuff and ADE20K datasets, demonstrating its strong generalization ability. The lower ΔCF indicates more precise colorization results, and we achieve the lowest ΔCF across all datasets, suggesting its effectiveness in achieving natural and lifelike colorization results. Furthermore, the best scores on PSNR demonstrate

MultiColor colorizes images with plausible colors. MultiColor outperforms all previous work with significant margins.

**Qualitative Comparison.** Figure 5 presents visualization of image colorization results. We display comparisons of images in different scenes from the ImageNet validation dataset. Note that the GT images are provided for reference only but the evaluation criterion should not be color similarity. A noticeable trend is that our results exhibit a more vivid appearance. We can see that the ball colorization (Row 1) of previous methods looks unnatural and introduces color bleeding effects in contrast to our consistent dark blue. Meanwhile, our method produces saturated results for the hue of the background (lawn). InstColor [36] employs a pre-trained detector to detect objects and cannot color the whole image well (Column 2). GCPColor [44] and ColorFormer [17] usually lead to incorrect semantic colors and low color richness. DDColor [18] may produce rare colors. Instead, our method maintains the consistent color and captures the details as shown in row 2, 3 and 4 of Figure 5. Furthermore, our method can yield more diverse and lively colors for whole image as shown in the last row of Figure 5.

## 5.3 Ablation Study

We now perform a series of ablation studies to analyze MultiColor. We first verify the effectiveness of modeling multiple color spaces strategy, and then study the effect of some other settings about the CSCNet. Finally, we investigate the effect of the feature scales. All ablation results are evaluated on the ImageNet val-5k dataset.

**Color Space.** Table 3 shows the effectiveness of the strategy for modeling multiple color spaces. We can observe that the incorporation of multiple color spaces significantly facilitates colorization performance. Additionally, we visualize the colorized results of the ablation experiments in Figure 6. In single color space, the overall color of the colorized images may be incorrect semantic colors and low color richness. The combination of multiple color spaces makes the restored colors more realistic and higher saturation. For example, the color of the chicken is more consistent with human perception (Row 1). Such a remarkable promotion benefits from complementarity among multiple color spaces, which can capture not only saturation of hues but also color contrast.

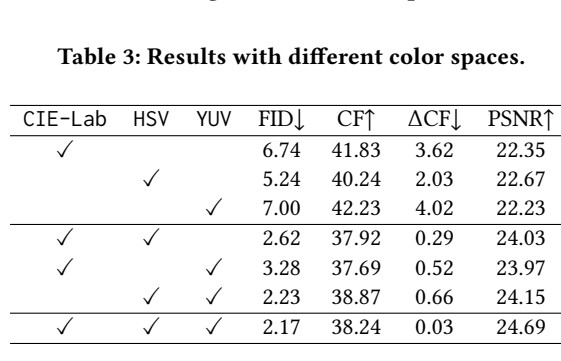

|       |       |       |        |       |        |            |
|-------|-------|-------|--------|-------|--------|------------|
| Input | InstColor | GCPColor | ColorFormer | DDColor | Our method | Groundtruth |

Figure 5: Visual comparison of competing methods on automatic image colorization.

Table 3: Results with different color spaces.

| CIE-Lab | HSV | YUV | FID↓ | CF↑ | ΔCF↓ | PSNR↑ |
|:---:|:---:|:---:|:---:|:---:|:---:|:---:|
| ✓ |   |   | 6.74 | 41.83 | 3.62 | 22.35 |
|   | ✓ |   | 5.24 | 40.24 | 2.03 | 22.67 |
|   |   | ✓ | 7.00 | 42.23 | 4.02 | 22.23 |
| ✓ | ✓ |   | 2.62 | 37.92 | 0.29 | 24.03 |
| ✓ |   | ✓ | 3.28 | 37.69 | 0.52 | 23.97 |
|   | ✓ | ✓ | 2.23 | 38.87 | 0.66 | 24.15 |
| ✓ | ✓ | ✓ | 2.17 | 38.24 | 0.03 | 24.69 |

Table 4: The impact of the block number in CSCNet. $L_1$ to $L_4$ corresponds to the layer numbers as in Table 1.

| $[L_1, L_2, L_3, L_4]$ | FID↓ | CF↑ | ΔCF↓ | PSNR↑ |
|:---:|:---:|:---:|:---:|:---:|
| $[1, 0, 0, 0]$ | 4.35 | 40.33 | 2.12 | 23.07 |
| $[1, 1, 0, 0]$ | 3.64 | 39.51 | 1.30 | 23.55 |
| $[1, 1, 1, 0]$ | 2.43 | 38.54 | 0.33 | 24.47 |
| $[1, 1, 1, 1]$ | 2.17 | 38.24 | 0.03 | 24.69 |

**CSCNet Settings.** We use stack of blocks for the complementary network as shown in Table 1. The relationship between the number

of the blocks and the colorization performance is presented in Table 4. Fewer blocks have a limited capability to colorize grayscale images. When we increase the number of blocks step-by-step, the performance is boosted on all metrics.

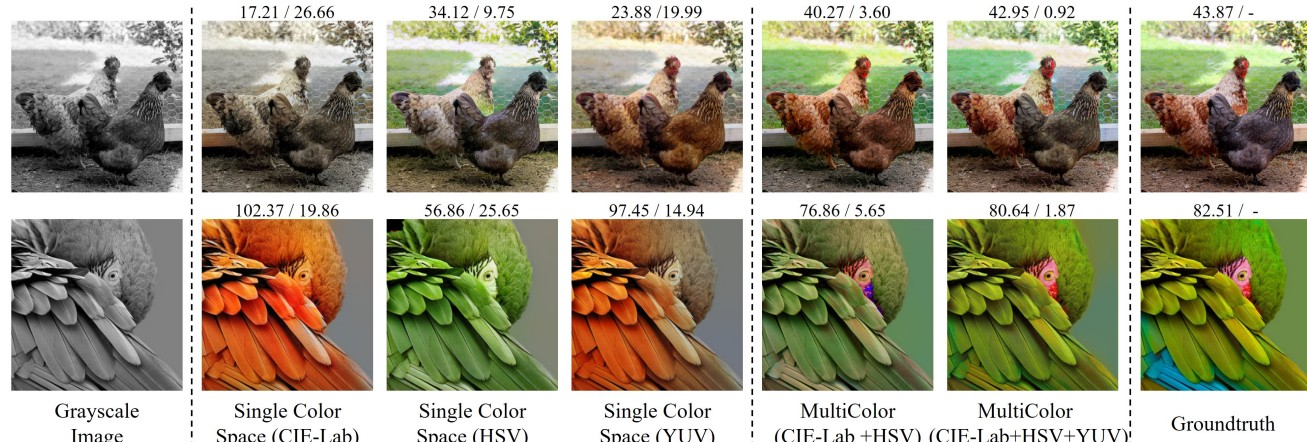

Figure 6: Visualization results by learning from different color spaces. The numbers on top of each image indicate CF / ΔCF.

**Table 5: Different kernels in CSCNet.**

| conv$_1$, conv$_2$ | FID↓ | CF↑ | ΔCF↓ | PSNR↑ |
|---|---|---|---|---|
| $1 \times 1, 1 \times 1$ | 2.28 | 38.36 | 0.15 | 24.49 |
| $3 \times 3, 3 \times 3$ | 2.21 | 38.27 | 0.06 | 24.56 |
| $5 \times 5, 5 \times 5$ | 2.23 | 38.30 | 0.09 | 24.53 |
| $1 \times 1, 3 \times 3$ | 2.17 | 38.24 | 0.03 | 24.69 |
| $1 \times 1, 5 \times 5$ | 2.21 | 38.28 | 0.07 | 24.62 |

**Table 6: Effectiveness of different feature scales.**

| | Feature Scales | FID↓ | CF↑ | ΔCF↓ | PSNR↑ |
|---|---|---|---|---|---|
| (a) | $F_1$ | 4.25 | 37.28 | 0.93 | 23.47 |
| | $F_2$ | 3.67 | 37.52 | 0.69 | 23.86 |
| | $F_3$ | 3.14 | 37.87 | 0.34 | 24.25 |
| (b) | $F_1 + F_2$ | 2.81 | 37.94 | 0.27 | 24.35 |
| | $F_2 + F_3$ | 2.56 | 38.03 | 0.18 | 24.47 |
| | $F_1 + F_2 + F_3$ | 2.17 | 38.24 | 0.03 | 24.69 |

We also analyze the influence of the kernel size of the blocks. As shown in Table 5, we can see that convolutional kernels of different sizes have an effect on performance. We choose the convolutional operation of $1 \times 1$ and $3 \times 3$ as it achieves the best results.

**Feature Scales.** Recall that the multi-scale features are generated from the upsampling operations of encoder. As shown in Table 6(a), we list the metrics generated by performing the framework on a single-scale feature only. We can observe that a feature with higher resolution tends to boost performance. Table 6(b) shows the results of our approach from multi-scale feature maps. It verifies that the multi-scale features for modeling color space is beneficial for image colorization. Figure 7 shows the visualization of features scale.

**Number of Transformer Decoder Operation.** We vary the number of transformer decoder operation $\frac{L}{3}$ to evaluate its effectiveness. The number is set to 3 in our default settings. Other operation number are also tried and the results are shown in Table 7. With the increase of the number, FID decreases from 4.22 to 2.17, and the PSNR increases from 23.77 to 24.69 dB.

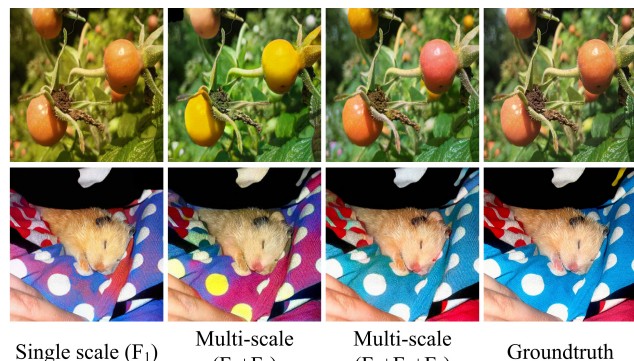

Single scale ($F_1$) · Multi-scale ($F_1+F_2$) · Multi-scale ($F_1+F_2+F_3$) · Groundtruth

Figure 7: Visualization results of ablation on feature scales.

**Table 7: Results with different number of the Transformer decoder operation.**

| Number | FID↓ | CF↑ | ΔCF↓ | PSNR↑ |
|---|---|---|---|---|
| 1 | 4.22 | 36.14 | 2.07 | 23.77 |
| 2 | 3.06 | 37.56 | 0.65 | 24.33 |
| 3 | 2.17 | 38.24 | 0.03 | 24.69 |

## 6 CONCLUSION

In this paper, we propose a novel framework for image colorization by learning from multiple color spaces. MultiColor combines various color information of multiple color spaces, for powerful and robust representation competence. Specifically, Transformer decoder progressively refines color query embeddings by leveraging multi-scale image features of encoder, followed by producing color channels of various color spaces through color mapper. Furthermore, we propose color space complementary network to combine color channels and achieve internal color space information complementation. Extensive experiments indicate that MultiColor achieve satisfactory colorization results.

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
