# OpenReview forum: "MultiColor: Image Colorization by Learning from Multiple Color Spaces"
_acmmm.org/ACMMM/2024/Conference — MM2024 Poster_

### Official Review · Reviewer_Kjth · 2024-05-23

**Rating:** 4
**Confidence:** 3

**Summary:**

The paper proposes a new approach called MultiColor for image colorization that leverages complementary information from multiple color spaces. It contains an encoder to extract multi-scale features, multiple colorization modules to model different color spaces separately, and a color space complementary network to fuse color channels from different spaces. The colorization modules employ a transformer decoder to refine learnable color queries and predict color channels, while the CSCNet generates the final colorized image.

**Strengths:**

1.	The use of multiple color spaces for image colorization seems novel and well-motivated.
2.	The modular design with dedicated colorization blocks for different color spaces is clear and reasonable.
3.	The transformer-based architecture enables modeling long-range dependencies in the colorization task.

**Limitations:**

1.	The motivation for using multiple color spaces could be strengthened with more qualitative/quantitative analysis of their individual characteristics and complementarity. It is somewhat surprising that adopting such a simple design can bring so much improvement. More insights into how the multi-color space strategy helps could be provided, such as visualizing intermediate results or predicted channels.
2.	The design of CSCNet is simple but its effectiveness needs more discussion and analysis through ablation studies.
3.	How do individual colorization modules handle color space transformations internally? More details are needed.
4.	Will the performance improve or degrade with increasing number of color spaces? An analysis is required. It seems that HSV space is the most useful one.
5.	For each color channel module, is the supervision information separate?
6.	Since this method adopts multiple color spaces with multiple networks, the complexity and parameters may be larger than other methods. Detailed complexity comparison is needed.
7.	More qualitative results and user study are recommended.

**Suitability:**

2

---

### Official Review · Reviewer_vf7x · 2024-05-24

**Rating:** 4
**Confidence:** 2

**Summary:**

This paper focuses on the image colorization task and propose MultiColor, a learning-based approach to colorize grayscale images that combines clues from multiple color spaces. Both quantitative evaluation and visual samples are provided.

**Strengths:**

- The proposed method is intuitive and reasonable.
- The evaluation is sufficient and convincing.
- The implementation details are provided and the ablation experiments are well conducted.

**Limitations:**

- It would be better if failure cases could be provided.
- The method proposed in this paper reconstructs color images directly from monochrome images without receiving any additional guidance. This greatly limits the application of the method in this paper, making it difficult to migrate to tasks such as image color editing.
- Comparisons with generative model-based approaches, such as L-CAD, are missing

**Suitability:**

3

---

### Official Review · Reviewer_ECa9 · 2024-05-27

**Rating:** 1
**Confidence:** 3

**Summary:**

This paper introduces a method called MultiColor, which utilizes the complementarity of multiple color spaces for image colorization. Initially, ConvNeXt networks are employed to obtain four different-sized feature maps. Subsequently, specialized colorization modules are used for each color space. Inspired by the success of query-based methods, the colorization modules for specific color spaces also incorporate powerful attention mechanisms combined with a series of learnable queries. Finally, a Color Space Complementary Network is introduced to integrate various color information from multiple color spaces.

**Strengths:**

A simple yet effective Color Space Complementary Network is designed to integrate various color information from multiple color spaces, enabling the acquisition of more color information from grayscale images in multiple color spaces.

**Limitations:**

1. Although the results of the ablation experiments indicate some complementarity among multiple color spaces, the paper lacks detailed theoretical analysis regarding the complementarity of different color spaces.

2. The Colorfulness metric results are not satisfactory, and the visualization results are average.

3. In the Color Space ablation experiments, there are doubts about whether the model was retrained when using only one or two channels. If the model was not retrained, the results in Table 3 cannot conclusively demonstrate the effectiveness of the color space model. (Because originally the model takes input from three color spaces for colorization, now with one color space missing, the colors may become less vibrant.)

4. The description of the content in the paper is not clear enough and requires further elucidation. For example, in Section 4.3, how are the results obtained from multiple color spaces {y_1, y_2, ..., y_n} fused? Is it acceptable to have a variable number of inputs (n)?

**Suitability:**

3

---

### Official Review · Reviewer_rcyN · 2024-06-02

**Rating:** 4
**Confidence:** 4

**Summary:**

This paper proposes to colorize the grayscale image by learning the clues from multiple color spaces. The transformer decoder and the color mapper modules are employed for each color space to extract the color channel prediction. The complementary network is used to generate the final colorized images. The results and the comparisons demonstrate the good performance of the proposed method.

**Strengths:**

The proposed framework extracts the various color channels of multiple color spaces and generates visually reasonable colorized images, which achieve better colorization results than previous methods, qualitatively and quantitatively.

**Limitations:**

1. What’s a failure case? I did not see any discussion about the failure or bad colorized results.
2. What's the inference time comparison, training cost comparison, or computation flops comparison? If the proposed model has a much larger complexity than previous methods, it may be unfair to compare the performance with previous methods.
3. Why not add a branch for RGB color space?
4. What is the ground truth image for Figure 1?

**Suitability:**

3

---

### Meta-Review · Area_Chair_zojD · 2024-07-01

**Recommendation:** Accept (Poster)
**Confidence:** 4

**Metareview:**

this paper investigates the colorization of grayscale images. specifically, it learns from multiple complementary color spaces using a transformer architecture and a complementary color module.

initially, the paper receives ratings of BA, R, BA, BA, with praises on the experimental results, the method being simple and effective, and the design being intuitive and well motivated. there are also concerns on the failure cases, reasoning behind system designs, lack of theoretical analysis, need for further clarification, and need for more discussion on the working mechanism. during rebuttal, the authors successfully addressed some concerns, and the final ratings improved to WA, R, BA, BA.

given the circumstances, the AC recommends Accept. although theoretical analysis would certainly help the readership, as the current manuscript has introduced its approach in a relatively clear manner and has successfully verified its effectiveness via experiments, the AC believes this is worth sharing with the community. with that said, the authors are strongly encouraged to include the content requested by reviewers (e.g. failure cases, discussion on the working mechanism) in their final version.